# Chronic Kidney Disease-Associated Pruritus

**DOI:** 10.3390/toxins13080527

**Published:** 2021-07-28

**Authors:** Puneet Agarwal, Vinita Garg, Priyanka Karagaiah, Jacek C. Szepietowski, Stephan Grabbe, Mohamad Goldust

**Affiliations:** 1Department of Dermatology, SMS Medical College and Hospital, Jaipur 302004, Rajasthan, India; doc.puneetagarwal@gmail.com; 2Consultant Nephrologist, MetroMas Hospital, Jaipur 302020, Rajasthan, India; doc.vinitagarg@gmail.com; 3Department of Dermatology, Bangalore Medical College and Research Institute, Bangalore 560002, Karnataka, India; pri20111992@gmail.com; 4Department of Dermatology, Venereology and Allergology, Wroclaw Medical University, 50-367 Wroclaw, Poland; jacek.szepietowski@umed.wroc.pl; 5Department of Dermatology, University Medical Center Mainz, Langenbeckstraße 1, 55131 Mainz, Germany; stephan.grabbe@unimedizin-mainz.de

**Keywords:** chronic kidney disease, pruritus, end-stage renal disease

## Abstract

Pruritus is a distressing condition associated with end-stage renal disease (ESRD), advanced chronic kidney disease (CKD), as well as maintenance dialysis and adversely affects the quality of life (QOL) of these patients. It has been reported to range from 20% to as high as 90%. The mechanism of CKD-associated pruritus (CKD-aP) has not been clearly identified, and many theories have been proposed to explain it. Many risk factors have been found to be associated with CKD-aP. The pruritus in CKD presents with diverse clinical features, and there are no set features to diagnose it.The patients with CKD-aP are mainly treated by nephrologists, primary care doctors, and dermatologists. Many treatments have been tried but nothing has been effective. The search of literature included peer-reviewed articles, including clinical trials and scientific reviews. Literature was identified through March 2021, and references of respective articles and only articles published in the English language were included.

## 1. Introduction

Pruritus is a distressing comorbidity found commonly in patients with end-stage renal disease (ESRD), chronic kidney disease (CKD), and those on dialysis.It adversely affects the quality of life (QOL) of these patients. The pruritus associated with CKD has been termed “uremic pruritus”; however, since there is no direct relationship between pruritus and uremia, the term “CKD-associated pruritus” (CKD-aP) is preferred [1]. The lack of specific clinical features defining the disease and consensus guidelines on diagnosis and management has lead to under-diagnosis and poor patient management. In a large international dialysis outcome and practice pattern study (DOPPS), conducted from 1996 to 2015, comprising of 51,062 patients from 21 countries, medical directors in 69% of the facilities under-diagnosed pruritus. Almost 17% of patients having itchy skin did not mention this symptom [2]. Thus, there is a need for a better understanding of this condition to provide efficient care for such patients. This article reviews the prevalence, pathogenesis, diagnosis, and management of CKD-aP.

## 2. Epidemiology

The prevalence of CKD-aP has been very variable in different studies. It has been reported to range from 20% to as high as 90% [2,3,4]. The prevalence has been reported to vary significantly in different countries and also within a country at different centers [5]. However, the prevalence has been observed to decline due to the availability of better dialysis facilities, different dialysis modalities, access, and adequacy. The prevalence also depends on the mode of dialysis, i.e., hemodialysis (HD) or peritoneal dialysis (PD). A large international dialysis outcome and practice pattern study (DOPPS) was conducted on hemodialysis (HD) patients from 12 countries [4]. It was completed in two phases, DOPSS I (1996–2001) with 17,034 patients and DOPPS II (2002–2004) with 12,839 patients. 45% of patients had moderate to severe pruritus in DOPPS I and 42% in DOPPS II. In DOPPS phase V (2012–2015), it was observed that the number of patients with pruritus declined between 1996 and 2015 from 28% to 18% [2]. Another large dialysis outcome and practice pattern study (DOPPS) was conducted in Japan. It was completed in three phases, JDOPSS I (1999–2001; 65 facilities, 2757 patients), JDOPPS II (2002–2004; 60 facilities, 2286 patients), and JDOPPS III (2005–2008; 62 facilities, 2326 patients). The prevalence of moderate to extreme pruritus was 44%, while those bothered by itching to some extent was 35% [5]. In a large American study, the prevalence of CKD-aP was 60% [6]. In a study on 382 patients in China, 51.7% of patients were reported to have CKD-aP, with a higher incidence in patients on HD than PD [7]. In a meta-analysis of cross-sectional studies, including 42 studies, the prevalence of CKD-aP in each study ranged between 18% and 97.8%, and the overall prevalence of CKD-aP was 55%. The pooled prevalence in males and females was similar (55%). The prevalence of pruritus in patients undergoing HD was similar to those undergoing PD (55% vs. 56%) [8]. The prevalence was found to depend on the method of dialysis and, thus, is different in HD and PD patients. In a cross-sectional study from Korea, the prevalence of mild pruritus was higher in PD patients (62.5%) than in HD patients (48.3%). There were 223 PD and 425 HD patients enrolled in the study. The VAS score in PD patients was found to be significantly higher than in HD patients [9]. However, in another study, the pruritus prevalence and VAS score was higher in HD (38.2%) than PD (28.6%). This significantly lower intensity of pruritus in PD patients was attributed to better preservation of renal function and less inflammation in patients on PD than HD [10]. Thus there is no consensus as to which dialysis method is better to prevent CKD-aP. In a few studies, the prevalence was determined in CKD patients who had not yet started dialysis. In a cross-sectional study on patients with stage 2–5 CKD, 18.9% had pruritus [11]. They found no relationship between pruritus and the stage of CKD. However, in another cross-sectional study of 200 CKD and ESRD patients, pruritus was seen in 36% of cases, and the authors found raising skin manifestations with the worsening of kidney disease [12]. In another study, 56% of 55 patients with stage 4 and 5 CKD reported pruritus [13]. A study on 5658 patients with CKD stage 3–5 (nondialysis) reported moderate to extreme pruritus in 24% of patients, across all stages. In these patients, other features observed were moderate-to-extreme dry skin (73%) and restless sleep for at least 3 days out of the week (48%), which was 19% and 26%, respectively, for patients without pruritus [14]. Pediatric patients on dialysis seldom encounter severe pruritus. In a systematic review involving 199 children at various German pediatric dialysis centers, only 9.1% children had pruritus, and even if present, was not severe [15]. In another multicenter study in Poland, 20.8% of the pediatric population with CKD experienced pruritus. It was noted that the children on dialysis had higher rates of pruritus (23.5%) compared to those on conservative supportive care without dialysis (18.4%). Children with pruritus were more commonly associated with xerosis (66.7%) than those without (50.9%) [16]. Some literature reviews on CKD-aP report that kidney transplant seems to relieve pruritus [17]. In a study on 49 patients with successful kidney transplantation, it was observed that when sufficient renal function was restored after transplantation, the uremic skin alterations disappeared [18].

## 3. Etiopathogenesis

The pathogenesis of CKD-aPhas not been completely understood. Many theories have been proposed in various studies to explain it. The most common of them are highlighted in Figure 1:

### 3.1. Immune Dysregulation

There are a lot of factors that suggest that CKD-aP is a systemic inflammatory disease. The dysregulation of the immune system and an increase in proinflammatory mediators may lead to CKD-aP. The beneficial effect of certain immunomodulators (ultraviolet B, tacrolimus, thalidomide, etc.) in uraemic pruritus supports this hypothesis. Th1 cells, serum C-reactive protein (CRP), interleukin (IL)-6, and IL-2 levels have been found to be significantly raised in these patients, further supporting the significance of inflammation in CKD-aP [19]. In the DOPPS study, a number of factors mediating immune function or inflammation were found to be associated with moderate to extreme pruritus, such as WBC count >6700 WBC/mL, hepatitis C, or ascites [4]. Recent studies have highlighted the role of interleukin 31 in the development of pruritus in chronic CKD patients. Interleukin 31 was significantly elevated in patients undergoing HD compared to the control group [20,21].

### 3.2. Xerosis

Xerosis manifests as dry and scaly skin. It has been reported to be a major feature in patients with pruritus [7]. The dysfunction of the sebaceous glands and apocrine sweat glands are the main factors leading to xerosis. An increase in the pH of the stratum corneum and increased vitamin A concentration also contribute to xerosis. The increase in skin pH may lead to disturbance in the activation of proteases involved in stratum corneum desquamation. Thyroid hypoactivity and mast-cell-mediated chronic inflammation may also contribute to skin xerosis [22]. This dry skin causes a defective skin barrier function and accumulation of metabolites in the skin. These, along with the increased secretion of urea in sweat, leads to pruritus in CKD patients. In a study on 5658 patients with CKD stage 3–5 managed without dialysis, the authors found that those with severe pruritus had dry skin as compared to others [14]. However, a cross-sectional study completed on 167 HD patients in Iran did not find any difference in the prevalence of xerosis between patients affected with pruritus and those without [23]. The epidermal moisture content, according to a study, was significantly decreased in patients with CKD-aP, especially in those on HD. Cutaneous perfusion studies have reported dermal dehydration following hemodialysis due to fluid shift, which may be the cause for xerosis in CKD-aP. The study also found increased transepidermal water loss in these patients, suggesting the presence of impaired epidermal barrier in CKD [24]. Similar results were seen in children affected with CKD and undergoing dialysis [25]. The components of cornified envelope such as free fatty acids, cholesterol, cholesteryl esters, ceramides (1–3), triglycerides, and squalene were significantly reduced in patients with CKD in comparison to the control group; however, there was no significant difference between patients with pruritus and without pruritus [23]. Thus, treatment with emollients such as glycerol and paraffin in CKD-aPis known to objectively reduce the thickness and density of the scales along with significant improvement in the pruritus score and quality of life of the patients [26].

### 3.3. Hyperparathyroidism

A low level of calcium promotes secondary hyperparathyroidism in patients with CKD. This leads to a higher secretion of parathyroid hormone (PTH), which regulates blood phosphorus levels. The levels of serum phosphorus, calcium phosphorus products, and PTH were found to be elevated in patients with CKD-Ap [4,5]. Total parathyroidectomy has been reported to relieve pruritus in patients with secondary hyperparathyroidism [7]. However, hyperparathyroidism is not present in all patients with CKD-aP. In some patients with high levels of PTH, parathyroidectomy does not lead to improvement in pruritus. Additionally, injecting PTH does not cause pruritus, suggesting that it has no primary role in activating itch fibers. Therefore, parathyroidectomy as a treatment for pruritus is effective only in those patients where hyperparathyroidism is a major factor for CKD-aP [7].

### 3.4. Uremic Toxin Accumulation

Reduced renal function in patients with CKD causes inadequate excretion of metabolites. This leads to the accumulation of toxic metabolites causing various side effects. When the blood phosphorus levels are elevated beyond the normal limit, it combines with serum calcium to form calcium phosphate, which in turn gets deposited in the skin and other tissues. The calcium phosphate thus deposited activates local nerve fibers and causes pruritus [7]. In a study by Hsu et al., the authors found that theHD patients withhigher serum aluminum (Al) levels had a higher prevalence of CKD-aP and showed the importance of reduced exposure to Al in patients with pruritus [27]. Another study used ultraperformance liquid chromatography coupled with time-of-flight mass spectrometry (UPLCMS) for metabolic profiling of serum from CKD-aP patients to find diagnostic biomarkers for uremic pruritus, if any. They divided the patients into two groups according to the severity of pruritus and found nine metabolites (LysoPE (20:3(5Z,8Z,11Z)/0:0), p-cresol glucuronide, LysoPC(20:2(11Z,14Z)), hypotaurine, 4-aminohippuric acid, LysoPC(16:0), phenylacetic acid, kynurenic acid, and androstenedione) to be responsible for severe CKD-aP [28].

### 3.5. Neural Dysfunction

A role of deranged neural function has also been suggested as a possible mechanism for CKD-aP. In general, pruritus may be caused due to some centrally acting mediators that do not damage the central nervous system but trigger the itch pathway. The defect may be in the peripheral sensory pathway, or there might be a cortical hypersensitivity, decreased cortical inhibitory mechanism, or a defective spinal cord inhibition. Sorouret al. measured the levels of neurotropins, a group of neurological mediators causing pruritus in skin (brain-derived nerve growth factor (BDNF), neurotrophin-4 (NT-4)) in CKD-aP patients. NT-4 was elevated in uremic patients with a positive correlation with the severity of pruritus. Serum BDNF levels were higher in uremic patients than the controls, but there was no significant difference in levels between uremic patients with pruritus vs. those without pruritus. The authors hypothesized that these neurotrophinsmightfacilitate the function of other mediators of pruritus [29]. Serum levels of B-type (brain) natriuretic peptide (BNP), an itch-selective neuropeptide found in pruriceptive neurons of mice, were found to be frequently elevated in hemodialysis patients. In a cross-sectional study by Shimizu et al., the authors concluded that a higher level of serum BNP leads to an increase in CKD-aP, predominantly in daytime [30]. The expression of ion channels present in the skin was investigated by Momoseet al. They also compared the ion channels present in peripheral nerve endings between CKD patients with and without CKD-aP. They found a significantly elevated expression of Cav3.2, BKCa, and anoctamin1, and reduced expression of TRPV1 in CKD-aP patients. They concluded that these changes in expression of ion channels in CKD patients might increase generator potential related to itching [31].

### 3.6. Histamine Mechanism

Elevated levels of mast cells were found in the dermis of patients with CKD-aP [32]. Histamine is a mediator of mast cells and a potent elicitor of itch. Another mediator of mast cell is tryptase, which also mediates itch. Studies have shown raised levels of tryptase, histamine, and eosinophils in patients with CKD-aP. However, antihistaminics are relatively ineffective in treating CKD-aP [33,34].

### 3.7. Opioid Mechanism

An important mechanism for CKD-aP involves endogenous opioid peptides as well as the opioid system. Pruritus has been found to be induced by the stimulation of the μ-opioid system. It is supported by the fact that morphine (μ-opioid agonist) triggers itch. On the other hand, stimulation of the κ-opioid system reduces itching. Thus, it was concluded that the substance P-induced itch is inhibited by κ-opioid receptor stimulation and antagonism of μ-opioid receptors [3,35]. Studies have supported this theory as there was a significant reduction in κ -opioid receptor expression (*p* < 0.02) in the skin of patients with CKD-aP compared to the control group without an itch. A significant negative correlation was established between the intensity of pruritus and κ -opioid receptor expression (*p* = 0.002) [36].

### 3.8. Other Reported Hypothesis

Proteinase-activated receptors (PARs) are G-protein-coupled receptors activated by certain proteinases. Amongst them, PAR-2, cleaved by trypsin-like serine proteases, is associated with acute inflammation. PAR-2 is a histamine-independent mediator of itching. In a study by Moon et al., the expression of epidermal PAR-2, especially in the upper epidermis, was significantly higher in ESRD patients as compared to healthy controls. In situ zymography also confirmed these observations [37]. A cross-sectional study by Huang et al. indicated that environmental nitrogen dioxide (NO2) and carbon monoxide (CO) levels are significant factors contributing to CKD-aP and emphasized the importance of the living environment in dialysis patients [38].

## 4. Risk Factors Associated with CKD-aP

There are various risk factors associated with CKD-aP. Many studies have evaluated these factors and have reported variable results. In an international prospective cohort study on HD patients, the patients with pruritus were older, had higher median C-reactive protein, and had hepatitis B or C antibodies and low serum albumin. However, no association was observed with parathyroid hormone (PTH), Kt/V, serum calcium, phosphorus, calcium phosphorus product, and hemodiafiltration [2]. DOPPS data suggested that the likelihood of moderate to extreme pruritus was more in male patients or patients with a history of systemic diseases involving the heart, lungs, central nervous system, or liver. Higher serum phosphorus (>5.5 md/dL) or serum calcium (>10.2 mg/dL) levels, higher calcium phosphate (>80 mg^2^/dL^2^), lower serum or white blood cells >6700 /mL were also independent risk factors. The following factors were found to be negatively associated with pruritus: early ESRD (3 months or less), or prolonged ESRD >10 years, or serum ferritin concentration ≥400 ng/mL. It was also observed that an increase in dialysis dose improved pruritus [4]. In a Japanese DOPPS (JDOPPS) study, some additional risk factors were identified for CKD-aP, such as: hypertension, high PTH levels, smoked within the past year, and arteriovenous graft as the primary vascular access. They did not find any relation of CKD-aP with hepatitis B, serum creatinine, body mass index, dialyzer membrane type or flux of the dialyzer, percentage transferrin saturation, neutrophil count, and residual renal function [5]. In a study on 5658 patients with CKD stage 3–5 managed without dialysis, the following patient characteristics were associated with a higher prevalence of moderate-to-extreme pruritus: older age, female sex, history of lung disease and diabetes, higher serum phosphate, and lower hemoglobin levels. The prevalence was higher in patients with advanced stages of CKD [14]. Ozenet al. from Turkey found inflammatory mediators to be elevated in patients with CKD-aP. They found that WBC counts ≥ 6.7 × 10^3^/μL increased the risk of pruritus by 1.73 times. Serum proinflammatory cytokines (IL-6, CRP, etc.) were found to be higher in CKD-aP patients [39]. In a prospective cohort study of 85 PD patients in Taiwan, 28.2% of patients had pruritus. Technical faults in PD lead to a higher intensity of pruritus and worse patient survival in patients undergoing dialysis. They also found that higher blood levels of iPTH, higher dietary protein intake, long duration of dialysis, weekly total Kt/V ≤ 1.88, and high-sensitivity CRP were independent determinants of higher Visual Analog Scale scores of pruritus intensity [40]. A cross-sectional study was conducted in Korea amongst 223 PD and 425 HD patients. They observed that peritoneal dialysis and BMI were independently responsible for pruritus. The severity of pruritus was negatively correlated to total weekly Kt/V in PD patients and serum albumin levels in HD patients. In PD patients, it was positively correlated with duration of dialysis, total cholesterol levels, and systolic BP. Amongst all these factors, serum albumin levels in HD and total weekly Kt/V in PD were independently found significant. However, clinically, there was no significant difference between PD and HD patients in the distribution of pruritus, frequency of pruritus-related sleep disturbances, pruritus intensity, and scratching activity [9]. Ko et al. followed up 111 patients of maintenance hemodialysis with CKD-aP for 4 years. They compared the parameters in patients with improved and unimproved pruritus at the end of follow-up. Lower levels of calcium and phosphorus products, low serum phosphorus, and ferritin were seen in patients with improved pruritus. In the study, the factors responsible for increased severity of pruritus over time were: female gender, low uric acid at baseline, use of low-flux dialyzer, high urea nitrogen, and β2- microglobulin, lowKt/V at baseline, low baseline VAS score, total bilirubin at baseline, high creatinine, preprandial glucose, and high calcium and phosphate product at baseline. It was also observed that a baseline Kt/V˂ 1.5 had severe pruritus, while the use of a high-flux dialyzer led to improvement in pruritus [41]. A study by Tinghai Hu et al. did not find serum calcium levels to be associated with pruritus. In a meta-analysis of cross-sectional studies, dialysis duration was negatively associated with the prevalence of CKD-associated pruritus [7]. To summarize all these findings, low levels of serum albumin and high levels of serum C-reactive protein, phosphorus, PTH, and calcium phosphorous products are positively correlated with higher intensity of pruritus. Coinfection with Hepatitis C and white blood cells >6700 /mL also increases the intensity of pruritus. Dialysis efficiency also affects pruritus, and low Kt/V increases pruritus. Older age, female sex, and presence of comorbidities such as lung disease and diabetes mellitus are also poor prognostic factors.

## 5. Clinical Features

The pruritus in patients with CKD presents with diverse clinical features. However, the pruritus is generally persistent and recurrent, bilaterally symmetrical, and worse at night. It is almost present daily and mostly presents over the trunk and limbs, with the back more commonly affected. Heat and dryness seem to aggravate pruritus. There are generally no primary skin lesions associated with pruritus, but secondary lesions due to scratching may be seen, such as: excoriation, linear crusts, ulcerations, impetigo, papules and prurigonodularis [6]. Amongst 103 HD patients in the United States, almost 84% of patients had daily or nearly daily pruritus. The itching was present over a wide area which was discontinuous but bilaterally symmetric and may migrate over time. The itching was persistent, it was severe at night as compared to daytime, and was aggravated by showering, dialysis, heat, stress, cold, and physical activity [42]. In DOPPS, it was observed that almost 50% of patients experienced itching throughout the day with no specific time; however, one-third of patients were most bothered at night. The presence of itching was not associated with the timing of dialysis; however, itching was reported to be worse during dialysis in 15% of patients, soon after dialysis in 9% of patients, or on nondialysis days in 14% of patients [2]. In a study by Tinghai Hu et al., the authors found that in most patients, pruritus was symmetric, while some had generalized pruritus. The pruritus was persistent and recurrent. The majority of patients suffered from daily itching, while the rest had itching occasionally in a week or a month. There were no typical skin lesions; however, xerosis was commonly present. Heat and dryness were found to aggravate itching [7]. Ozenet al. reported pruritus to be generalized in 35.3% of patients, with 50.4% experiencing moderate pruritus amongst the 249 HD patients studied. A total of 39.1% of patients had severe pruritus on the day after dialysis. The mean pruritus severity was 6.47 ± 1.56. If the patient had dry skin, they were 0.194 times more likely to have very severe pruritus [39]. In a cross-sectional study on 167 HD patients from Iran, pruritus was generalized in 70%of patients. The pruritus was limited to the trunk and limbs in 14.3% each and to the head and neck in 1.4%. In patients with pruritus, neuropathy was significant in 63.8% of patients [23]. Minato et al. studied 46 PD patients and found that pruritus affected almost all parts of the body, with the back being the most common site (70%). The other parts involved were lower limbs (67%), chest and abdomen (59%), upper limbs (28%), and head and neck (22%). The intensity was higher in the night times, as shown by the VAS score [43]. A study from Turkey on 181 HD patients reported itching before the hemodialysis session in 86.7% of patients, during hemodialysis in 72.9% of patients, and after hemodialysis in 49.7% of patients. The majority of patients had itching for 6–12 h, whereas only 1.1% had itching throughout the day. The itch was moderate in 40.3%, mild in 30.4%, severe in 28.2%, and unbearable in 2%. The course of itching was “a little bit better, but still present” in 38.7%, while in 5%, it was “getting worse”, as explained by the patients.In 65.2% of cases 6–10 anatomical regions were involved, most commonly over the back followed by upper arm, chest, and abdomen. Palms, soles and face/head had very minimalinvolvement [44].

## 6. Diagnosis

The clinical presentation of CKD-aP is highly variable; the onset of symptoms varies amongst the patients and the intensity of pruritus may range from mild to severe. The itching frequency and timings also differ amongst them. Pruritus in CKD-aP may not be associated with any primary skin lesions, while in some patients, it may be associated with secondary skin lesions such as crusts, papules, ulcerations, erosions, impetigo, and prurigonodularis. It is a diagnosis of exclusion in the absence of other systemic conditions causing itch, such as:liver disorders, hematologic abnormalities, comorbid skin conditions, infestations, etc. [45].

The criteria for diagnosis of uremic pruritus is [45]:Pruritus appears shortly before the onset of dialysis, or at any time, without evidence of any other active disease that could explain the pruritus.More than or equal to three episodes of itch during a period of 2 weeks, with the symptom appearing a few times a day, lasting at least few minutes, and troubling the patient.The appearance of an itch in a regular pattern during a period of 6 months, but less frequently than listed above.

Since itching is a subjective phenomenon, it is typically accessed via patient-reported outcomes (PROs). Several PRO scales have been developed for the measurement of pruritus intensity, which can be either unidimensional (measures only severity of pruritus) or multidimensional (measures severity and other characteristics of pruritus, such as: duration, impact on activities of daily life, direction, degree, and location of pruritus). There are also few scales that focus predominantly on the quality of life [16]. A transformation equation has been suggested to correlate the data between the multidimensional (5-D itch scale) and unidimensional (numeric rating scale (NRS)) scales [46]. The commonly used scales have been listed in Table 1.

## 7. Quality of Life in CKD-aP

CKD-aP is very distressing to patients and thus has a significant effect on their quality of life (QOL). It can affect their sleep and social functioning. The patients with decreased QOL were found to have higher mortality. It can keep patients awake at night, causing them to feel drowsy during the day and thus not have enough sleep, further affecting their daily activities [3]. In DOPPS V, it was observed that 60% of patients with moderate to extreme pruritus had inadequate sleep. Patients were often: (1) bothered by the appearance of skin, (2) frustrated or annoyed by itching, (3) bothered by effects of itching on interactions or a desire to interact with others, and (4) bothered by itching that it becomes hard to work [2]. In DOPPS Iand II, the patients’ sleep quality was severely affected by the degree of pruritus. The odds of not having enough sleep, being awake at night, or feeling sleepy during the day was 1.4–4.0 higher in HD patients with moderate to extreme pruritus. The mortality risk in these patients was higher by 13% in DOPPS I, 21% in DOPPS II, and 17% in DOPPS I and II combined [4]. Similar results were found in the JDOPPS study, where moderate to extreme pruritus lead to poor sleep quality in 26% of patients. If the patient had extreme pruritus, there was a 1.9–3.7 times higher risk of having poorer sleep quality. Additionally, there was a 22% higher mortality risk in patients with moderate to extreme pruritus [5]. In a study by Ozen et al., 33.8% of patients had sleep disturbances due to pruritus [39].

## 8. Treatment

The patients with CKD-aP are mainly treated by nephrologists, primary care physicians, and dermatologists. In an international study, almost 50% of patients were treated by a nephrologist, 19% by a primary care physician, and only 24% were managed by a dermatologist [2]. Since there are many factors associated with CKD-aP, and these factors vary considerably amongst patients, no universal treatment can apply to all. The patients have to be properly evaluated, including proper history and examination, and the treatment planned accordingly. Mentioned below are various factors responsible for itching and methods to manage them.

### 8.1. Management of Xerosis

Frequent application of soap and hot water, whether in bathing or hand washing, should be avoided by the patient, as it may aggravate dryness. Emollients are mainly used to treat xerosis. There are no randomized control trials to suggest as to which emollient is best for CKD-aP; however, there are few studies on the use of various emollients. In a prospective open pilot trial, topical 10% urea plus dexpanthenol lotion was effective in reducing xerosis in dialyzed patients [47]. Another study evaluated the effect of an aqueous gel with a higher water content which was applied twice daily for 2 weeks and stopped for 2 weeks thereafter. The gel contained 20 g of naturally derived vitamin E, aloe vera extract, squalene, and other naturally derived ingredients and 80 g of water. The control group was not prescribed any emollient for 4 weeks. They concluded that in mild pruritus, this emollient could reduce itching and improve xerosis in comparison to the control group with no emollients [48]. A randomized, double-blind, intraindividual (left versus right comparison), multicentric clinical study was performed to evaluate an emulsion combining glycerol and paraffin on 99 patients with moderate to severe uremic xerosis. They observed that this test product was highly effective in 72 patients (73%), whereas 44 patients (44%) responded to the comparator [25].

### 8.2. Antihistamines

The effect of histamine can be antagonized by two methods: first, by histamine receptor antagonists, such as hydroxyzine, fexofenadine, etc., and second, by preventing histamine release by use of mast cell stabilizers, such as cromolyn sodium, montelukast, etc.Studies have found limited or no role of histamine receptor antagonists in relieving CKD-aP [3]. Ketotifen, a mast cell stabilizer was found to be effective in CKD-aP [34,49]. Cromolyn sodium, another mast cell stabilizer, was effective in decreasing pruritus. Its adverse effects reported are burning sensation and flatulence [50]. Nicotinamide, a potent stabilizer of mast cells and leukocytes, was not found to be effective, while oral zinc sulfate was effective in relieving pruritus [51].

### 8.3. Opioid Mediators

The µ receptor antagonist and κ- receptor agonist have been found to alleviate CKD-aP. The κ -opioid agonist additionally inhibits µ-receptor effects both centrally as well as peripherally. The drugs that target κ- opioid receptors (nalfurafine) and μ- opioid receptors (naltrexone) were reviewed in a systematic review and meta-analysis of randomized controlled trials. They found that both nalfurafine and naltrexone significantly reduced pruritus severity [35]. A randomized, double-blind, placebo-controlled study using nalfurafine in CKD-aP reported statistically significant improvement in “worst itching” VAS as compared to placebo after 2 weeks of nalfurafine therapy. The most common adverse effects observed were headache, nausea, insomnia, vertigo, vomiting, and elevations of liver enzymes, SGOT and SGPT, which were mild and resolved spontaneously, and did not lead to discontinuation of treatment [52]. Nalbuphine hydrochloride, a mixed μ-antagonist/κ-agonist opioid drug, has been found promising in treating CKD-aP. It has been tried both as a parenteral drug and as an extended-release tablet [53]. In a multicenter, randomized, double-blind, placebo-controlled trial, by Mathur et al., 373 CKD patients on hemodialysis with moderate or severe pruritus were randomized to receive either nalbuphine extended-release tablets 120 mg (NAL 120), or 60 mg (NAL 60), or placebo for 8 weeks. There was a reduction in the intensity of itching from the Numerical Rating Scale (NRS) of 6.9 by 3.5 in the group treated with NAL 120 mg and by 2.8 in the placebo groups. Patients on NAL 120 mg also experienced a reduction in sleep disruption due to pruritus (*p* = 0.062). However, NAL 60 mg showed no significant difference in comparison to placebo [54].

Difelikefalin (CR845), a peripherally restricted and selective κ-opioid agonist, was given for 8 weeks in HD patients with moderate to severe pruritus. Phase 3 randomized trial with 378 patients had 158 patients receiving difelikefalin, out of which 82 (51.9%) patients reported a reduction in the Worst Itching Intensity Numerical Rating Scale (WI-NRS) score by atleast 3 points compared to 51 (30.9%) of 65 patients receiving placebo. Difelikefalin also showed significant improvement in the quality of life from baseline to week 12 as evaluated by 5-D itch scale and the Skindex-10 scale. A total of 37.1% of the patients in the difelikefalin group showed a 4 point decrease in WI-NRS at week 12 compared to 17.9% in the placebo group (*p* < 0.001). Diarrhea, vomiting, and dizziness were more commonly reported in the difelikefalin group [55]. Another study demonstrated that difelikefalin reduced pruritus intensity and its duration. Sleep quality, itch-related QoL, including the ability to perform daily activities, mood/emotional distress, and social functioning, also improved with difelikefalin. The most frequently (≥5%) reported treatment-emergent adverse events were dizziness, diarrhea, somnolence, nausea, and fall [56].

### 8.4. Modulation of Neural Transmission

Peripheral C-fiber nerve transmission is blunted by drugs, such as gabapentin, pregabalin, capsaicin, and pramoxine. They act by preventing the release of neurotransmitters from presynaptic nerve terminals and thus, modulate itching and pain. These drugs have been found useful in CKD-aP [3]. Gabapentin, an antiepileptic agent, inhibits neuronal calcium influx and thus, modulates the neuronal pruritic sensation in uremia. In a systematic review on gabapentin, including seven studies, the author concluded that six of seven studies reported that gabapentin decreases pruritus intensity when used in a dose ranging from 100 to 400 mg post-HD or 900 mg daily orally. Another study recommended the use of gabapentin in CKD-aP if antihistamines and/or topical emollients were not effective. Adverse events reported in these studies were dizziness, fatigue, and somnolence. Due to neurological side effects, gabapentin has to be initiated at a low dose of 100 mg and gradually up-dosed depending on the response and tolerance of the patient, after consideration of renal parameters owing to its elimination by the kidneys [57]. Topical 6% gabapentin when compared with a vehicle showed a reduction in the CKD-aP severity with no acute adverse events after 2 weeks of treatment in the gabapentin group [58]. Pregabalin is another anticonvulsant useful in uremic pruritus [59]. Pregabalin is superior to gabapentin with respect to pharmacokinetics and pharmacodynamics. The oral absorption of gabapentin is slower than pregabalin. Furthermore the bioavailability of gabapentin reduces as its dose is increased. For pregabalin, the bioavailability remains more than 90% regardless of the dose [60]. Patients intolerant of gabapentin may tolerate pregabalin [58]. Capsaicin, a phytochemical (8-methyl-N-vanillyl-6 nonenamide), obtained from capsicum, can locally deplete neuropeptide substance *p* in the skin. It has been found useful in CKD-aP by a few authors [61,62]; however, a systematic review found no convincing evidence of the effect of topical capsaicin in pruritus [63]. Pramoxine, a topical local anesthetic, has been shown to improve pruritus when used both alone or as a combination with lactic acid. In a randomized, double-blind, controlled comparative trial, 1% pramoxine lotion reduced pruritus to a greater degree vs. placebo [64].

### 8.5. Uremic Toxin Removal

Kt/V has been found to be inversely related to CKD-aP [40]. Higher dialysis improves the prevalence and intensity of pruritus in hemodialyzed patients [65]. Neutral macroporous resin hemoperfusion (HP) was used in a study to treat maintenance hemodialysis patients (MHD) with refractory uremic pruritus. The authors concluded that HP was safe and effective in improving refractory pruritus and inflammation in MHD patients. Activated charcoal is a powerful, nonselective intestinal adsorbent. It has been found to relieve CKD-aP by removing uremic toxins [66,67]. Oral activated charcoal at a dose of 6 g/day could be used safely and is an effective, low-cost treatment in patients with uremic pruritus. However, nonselective adsorption of compounds can affect the bioavailability of certain drugs, and protein-bound toxins are not removed by charcoal. This can be overcome by hemoperfusion on activated charcoal in patients undergoing hemodialysis, which removes the protein-bound toxins with a favorableresult. Cartridges with activated charcoal can be used with hemodialysis. Oral charcoal is most frequently associated with side effects such as appetite loss, nausea, constipation, and gastrointestinal discomfort [67]. In a study, cholestyramine, a nonabsorbable anion-exchange resin, which binds organic acids intraluminally, has been found effective in relieving CKD-aP [65].

### 8.6. Phototherapy

The late 1970s marked the use of broadband UVB (BB-UVB) phototherapy for the first time in treating uremic pruritus.A total of 90% of patients treated with BB-UVB responded to therapy while as high as 80% remained free of itch after 7 months follow-up on an average. NB-UVB therapy for uremic pruritus was met with mediocre success, with 54% of patients showing improvement and 66% percent of the responders reported recurrence at 6-month follow-up [68]. In a systematic review analyzing phototherapy as a treatment for uremic pruritus, the author concluded that even though narrow-band-UVB(NB-UVB) is beneficial in decreasing pruritus and is less erythemogenic and carcinogenic than broad band-UVB (BB-UVB), BB-UVB is the treatment of choice for CKD-aP, while UVA had effects equivalent to placebo [68]. In a study on 30 patients from India, NB-UVB phototherapy was found to be effective in the treatment of intractable uremic pruritus [69]. NB-UVB was also found to be effective in CKD-aP associated with peritoneal dialysis [70].

### 8.7. Others

Gamma-linolenic acid in oral as well as topical formulations has been found to improve CKD-aP [71,72,73]. In a study, Omega-3 fatty acids were found to be more effective than placebo in decreasing uremic pruritus [74]. In a randomized crossover, double-blind trial, thalidomide caused a reduction in CKD-aP [75]. A similar observation was made in a review on thalidomide use in CKD-aP [76]. Turmeric, nicotinamide, sericin, pentoxifylline, nicergoline, doxepin, sertraline, ondansetron, tacrolimus, and ergocalciferol have all been found to relieve CKD-aP by their immunomodulatory effects [3]. In CKD-aP patients with secondary parathyroidism, parathyroidectomyhas been found to relieve pruritus [77,78]. Finally, acupuncture was foundto be useful in few studies [3]. A systematic review and meta-analysis found acupuncture and acupressure effective in uremic pruritus. Nociceptive sensation such as pruritus is transmitted by the large, unmyelinated, slow conducting C fibers from the periphery to the center. Acupuncture is capable of gating this sensation by the release of endogenous opiate-like substances that are carried by the smaller, myelinated, and rapidly conducting β and δ fibers that blunt the perception of itch peripherally and centrally [79]. Some studies have suggested that a vegetarian [80] and a low-protein diet [40] may help to reduce the intensity of CKD-aP.A renal transplant may prove to be helpful in patients suffering from chronic itch. In a questionnaire-based study, patients suffering from CKD for 20.2 ± 12.3 years were evaluated after a mean time of 8.0 ± 6.5 years. A total of 73.7% of cases reported complete cessation of itch after successful renal transplantation. A total of 42 (21.3%) renal transplant recipients reported itch, of which 22 (52.4%) cases reported the appearance of itch following transplantation. However, those with itching after renal transplant responded well to HD [81].

## 9. Conclusions

CKD-aP is a prevalent comorbidity associated with ESRD and CKD. It is still under-reported and undertreated by treating physicians. Many risk factors have been identified for it, and these factors, if carefully monitored and controlled, can lead to an improved quality of life in these already disease-burdened patients.

## Figures and Tables

**Figure 1 toxins-13-00527-f001:**
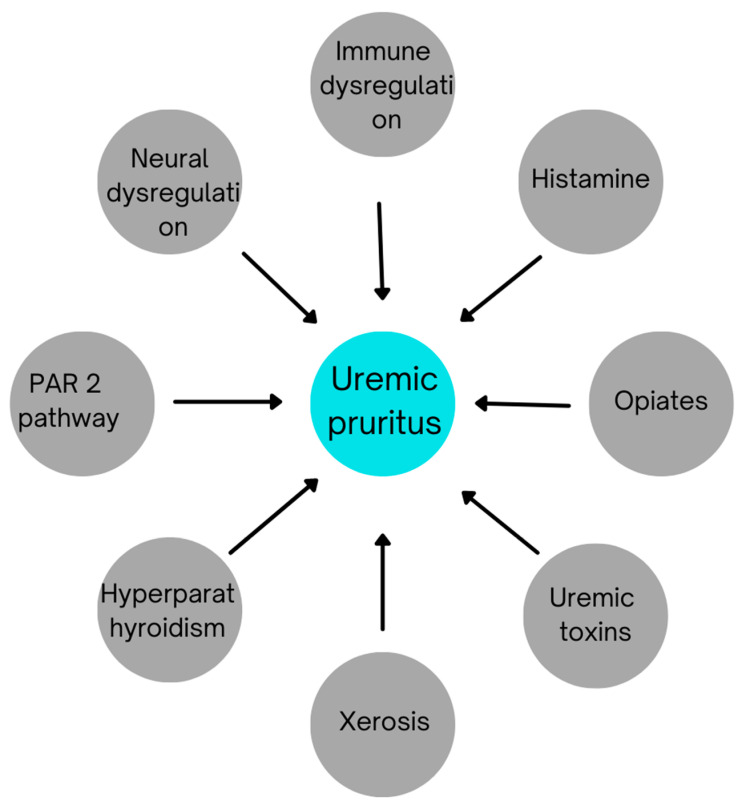
Etiological factors contributing to uremic pruritus.

**Table 1 toxins-13-00527-t001:** Symptom scales for evaluation and treatment of uremic pruritus [16].

Measurement	Tool
Unidimensional	Visual analog scale (VAS) Numeric rating scale (NRS) Verbal rating scale (VRS)
Multidimensional	Self-Assessed Disease Severity (ABC scale) Brief Itching Inventory Skindex-10 Itch MOS (Medical Outcomes Study) 5-D questionnaire Patient Benefit Index for pruritus (PBI-P) 4-Item Itch Questionnaire

## Data Availability

The data presented in this study are openly available.

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
