# Peer review of "Chronic Kidney Disease-Associated Pruritus"

_toxins, 2021, doi:10.3390/toxins13080527_

Round 1

Reviewer 1 Report

The authors described the pruritus in CKD progression and revealed the need for understanding and treating the symptoms. It's an interesting topic, however, there are several critical issues that have to be fixed before the manuscript is readable and meaningful. 

  1. ABSTRACT. At the end of the abstract, the author indicated the literature, or maybe the references were from MEDLINE and PUBMED. It's unnecessary sentences since the citation information can show the source.
  2. GENERAL. The citing style is hard to read. The English style needs to be improved by a native speaker. The manuscript is wordy.
  3. GENERAL. Some non-academic words in the manuscript need explain or replace with MeSH. e.g. advanced chronic kidney disease. Some words did not fit the manuscript. e.g. P1 L30, you are writing a review, and there is no subject or the hypothesis you need to 'examine'.
  4. GENERAL. The manuscript cited 83 references, however, no image or table to summarize or make the whole vision clear and tidy. As a novel disease or symptom, we need to know the organized vision at the very first step.
  5. INTRODUCTION. P1 L27. What's the meaning of 'the doctors finding it significant was less than 1%'?
  6. EPIDEMIOLOGY. The first part of this section is the same as those in the introduction and is unnecessary. 
  7. EPIDEMIOLOGY. There is no definition for the level of pruritus. In some sentences, the author described it as the 'moderate' and 'severe'. however, in the other sentences, the words turn out to be 'mild' and 'extreme'. Nobody knows how different between such words.
  8. EPIDEMIOLOGY. P2 L55, the author described the 55% vs. 56% as 'slightly lower, however, it is misleading to form the image that PD and HD are different in the prevalence. If there is no significance in the data analysis, please use neutral and objective words.
  9. EPIDEMIOLOGY. The author reported the clinical evidence in China and Korea is reversed in HD and PD patients with pruritus. As the same eastern Asian countries and almost same race, the author was failed to show the potential reasons for the difference.
  10. ETIOPATHOGENESIS. The author tried to reveal several possible mechanisms related to CKD and aP. However, there were no specific or novel points regarding the etiology. The author only showed some very common changes in CKD progression and only cited very few references. I suggest combining the etiology part to the previous section since the mechanism is unknown so far.
  11. ETIOPATHOGENESIS. In the 3.2 Xerosis section, the author only indicated several lipid-based NMF. Please also indicate several NMF in the other form (water-based and etc.)
  12. HYPERPARATHYROIDISM. Hyperparathyroidism also can be considered as the etiology evidence. The following semi-section is also not relevant to the main section.
  13. HYPERPARATHYROIDISM. The 4.3 and 4.4 is not the hypothesis since they are a well-known source of itch in living body.
  14. DIAGNOSIS. As a mechanistic unknown but important clinical issue, the diagnosis is critical and needs to discuss or review in detail. However, the author only cited very few and the frame is unclear and insufficient.
  15. PROGNOSIS. I think this part should be placed after the TREATMENT section.
  16. TREATMENT. It's not acceptable to see the histamine and opioid-related treatment since the author has declared they are only the hypothesis in the previous sections.
  17. TREATMENT. in the 9.4 semi-section, the author showed some drug/reagents. however, the clinical use of gabapentin and pregabalin is to suppress pain and only pramoxine is usually used for treating itch.
  18. TREATMENT. in 9.7 semi-section, the author introduced some novel or un-assigned therapeutic manners. However, one of the drugs, thalidomide has strong adverse effects which was not approved by FDA for a very long time. 
  19. As the conclusion part was in the TREATMENT section.

Author Response

Dear reviewers,

Thank you for your valuable insights and comments. Here are the responses to each of your concerns. 

1. Please describe more detailed information about the xerosis.

A: Duly noted and rectified

2. In line 339, how many times to use a moisturizing emollient is recommended? The emollient is a water-containing gel, what is the water content of it?

A: Details furnished as recommended

3. The article states that narrow band-UVB (NB-UVB) is beneficial in decreasing pruritus and is less erythemogenic and carcinogenic than broad band-UVB (BB- UVB). However, in the next sentence, it mentions BB-UVB is the treatment of choice for uremic pruritus. It does not seem logical. Please describe BB- UVB more detailed.

A: Detailed explanation given as per the comment.

4. Line 449, there are two types of nerve fiber conduction but no explanation what is spinal cord release of opioid-like substances.

A:Explanation provided.

5. At present, it is believed that gabapentin plays an increasingly important role in the treatment of uremic pruritus. In order to avoid neurological side effects, a small dose (100mg) should be started, and the dose should be adjusted according to renal function due to renal elimination.

A: Duly noted and modified accordingly

6. Please describe why and how oral charcoal control uremic pruritus.

A: Explanation provided

7. In the conclusion part, the pathogenesis of itching due to uremic toxicity has not yet been fully established, and its treatment methods have not been satisfactory so far. The best treatment method is combined treatment.

A: Comment taken into consideration and necessary changes are made.

Hope this answers all your queries. Hope to have the article accepted. 

Warm regards,

Authors

Reviewer 2 Report

Uremic pruritus is a common complication in patients with chronic kidney disease (CKD) and end-stage renal disease (ESRD). Although the incidence of uremiic pruritus has decreased with the advancement of hemodialysis and peritoneal dialysis technology in the past decade, but recently an international study, Dialysis Outcomes and Practice Patterns Study (DOPPS) counts more than 51,000 patients with chronic kidney disease. The results show that there is still a 69% incidence. It is believed that uremic pruritus is still a clinically important symptom of chronic kidney disease and affects patients. There are a number of conceptual issues that should be addressed by the authors to improve the clarity and quality of this manuscript.

Specific comments

  1. Please describe more detailed information about the xerosis.
  2. In line 339, how many times to use a moisturizing emollient is recommended? The emollient is a water-containing gel, what is the water content of it?
  3. The article states that narrow band-UVB (NB-UVB) is beneficial in decreasing pruritus and is less erythemogenic and carcinogenic than broad band-UVB (BB- UVB). However, in the next sentence, it mentions BB-UVB is the treatment of choice for uremic pruritus. It does not seem logical. Please describe BB- UVB more detailed.
  4. Line 449, there are two types of nerve fiber conduction but no explanation what is spinal cord release of opioid-like substances.
  5. At present, it is believed that gabapentin plays an increasingly important role in the treatment of uremic pruritus. In order to avoid neurological side effects, a small dose (100mg) should be started, and the dose should be adjusted according to renal function due to renal elimination.
  6. Please describe why and how oral charcoal control uremic pruritus.
  7. In the conclusion part, the pathogenesis of itching due to uremic toxicity has not yet been fully established, and its treatment methods have not been satisfactory so far. The best treatment method is combined treatment.

Minor comments

  1. Please provide an abbreviation of CKD-AP in line 409.
  2. Please correct chapter 10. to label“conclusion part”.
  3. Ref 82, the “vegetarian” of sentence should not refer to “acupuncture”.

Author Response

Dear reviewers,

Thank you for your valuable insights and comments. Here are the responses to each of your concerns. 

1. Please describe more detailed information about the xerosis.

A: Duly noted and rectified

2. In line 339, how many times to use a moisturizing emollient is recommended? The emollient is a water-containing gel, what is the water content of it?

A: Details furnished as recommended

3. The article states that narrow band-UVB (NB-UVB) is beneficial in decreasing pruritus and is less erythemogenic and carcinogenic than broad band-UVB (BB- UVB). However, in the next sentence, it mentions BB-UVB is the treatment of choice for uremic pruritus. It does not seem logical. Please describe BB- UVB more detailed.

A: Detailed explanation given as per the comment.

4. Line 449, there are two types of nerve fiber conduction but no explanation what is spinal cord release of opioid-like substances.

A: Explanation provided.

5. At present, it is believed that gabapentin plays an increasingly important role in the treatment of uremic pruritus. In order to avoid neurological side effects, a small dose (100mg) should be started, and the dose should be adjusted according to renal function due to renal elimination.

A: Duly noted and modified accordingly

6. Please describe why and how oral charcoal control uremic pruritus.

A: Explanation provided

7. In the conclusion part, the pathogenesis of itching due to uremic toxicity has not yet been fully established, and its treatment methods have not been satisfactory so far. The best treatment method is combined treatment.

A: Comment taken into consideration and necessary changes are made.

Hope this answers your query. Hope to have this accepted at the end of this.

Warm regards,

Authors

Round 2

Reviewer 1 Report

The authors added several updates, however, many of them need a clear citing source. Please add the citing information to the new-added evidence.

Author Response

Dear reviewer,

Most of the updates were extended from the existing references. However in view of your recent suggestions, few have been added and references are updated. Hope this has answered all your concerns. Hoping to have this manuscript accepted.

Warm Regards,

Authors